# *DMRT1* is a testis-determining gene in rabbits and is also essential for female fertility

Emilie Dujardin[1,2]*, Marjolaine André[1,2†], Aurélie Dewaele[1,2†], Béatrice Mandon-Pépin[1,2], Francis Poulat[3], Anne Frambourg[1,2], Dominique Thépot[1,2], Luc Jouneau[1,2], Geneviève Jolivet[1,2], Eric Pailhoux[1,2*‡], Maëlle Pannetier[1,2‡]

[1]Université Paris-Saclay, UVSQ, INRAE, BREED; 78350, Jouy-en-Josas, France; [2]École Nationale Vétérinaire d'Alfort, BREED; 94700, Maisons-Alfort, France; [3]Institute of Human Genetics, CNRS UMR9002 University of Montpellier; 34396, Montpellier, France

*For correspondence:
emilie.dujardin@inrae.fr (ED);
eric.pailhoux@inrae.fr (EP)

[†]These authors contributed equally to this work
[‡]These authors also contributed equally to this work

Competing interest: The authors declare that no competing interests exist.

**Abstract** DMRT1 is the testis-determining factor in several species of vertebrates, but its involvement in mammalian testes differentiation, where *SRY* is the testis-determining gene, remains ambiguous. So far, DMRT1 loss-of-function has been described in two mammalian species and induces different phenotypes: Disorders of Sex Development (46, XY DSD) in men and male infertility in mice. We thus abolished DMRT1 expression by CRISPR/Cas9 in a third species of mammal, the rabbit. First, we observed that gonads from XY *DMRT1⁻/⁻* rabbit fetuses differentiated like ovaries, highlighting that DMRT1 is involved in testis determination. In addition to SRY, DMRT1 is required in the supporting cells to increase the expression of the *SOX9* gene, which heads the testicular genetic cascade. Second, we highlighted another function of DMRT1 in the germline since XX and XY *DMRT1⁻/⁻* ovaries did not undergo meiosis and folliculogenesis. XX *DMRT1⁻/⁻* adult females were sterile, showing that DMRT1 is also crucial for female fertility. To conclude, these phenotypes indicate an evolutionary continuum between non-mammalian vertebrates such as birds and non-rodent mammals. Furthermore, our data support the potential involvement of *DMRT1* mutations in different human pathologies, such as 46, XY DSD as well as male and female infertility.

## eLife assessment

In this **important** study, the rabbit was used as a non-rodent mammalian model to show that DMRT1 has a testicular promoting function as it does in humans. The experiments are meticulous and **compelling**, and the arguments are clear and **convincing**. These results may explain the gonadal dysgenesis associated with mutations in human DMRT1 and highlight the need for mammalian models other than mice to better understand the process of gonadal sex determination in humans.

## Introduction

DMRT1 (Doublesex and Mab-3 Related Transcription factor 1) belongs to the highly conserved family of DM domain proteins, which exhibits a zinc finger DNA-binding motif that was initially identified in *Drosophilia* and *Caenorhabditis elegans* (**Erdman and Burtis, 1993**; **Raymond et al., 1998**). Some of its orthologs have been described as Testis-Determining Factor (TDF) in vertebrate species such as medaka (*Oryzias latipes*) (**Matsuda et al., 2002**), xenope (*Xenopus laevis*) (**Yoshimoto et al., 2010**), or chicken (**Smith et al., 2009**). In the last, the Z chromosome carries the *DMRT1* gene. In ZZ males, two

**eLife digest** Animals that reproduce sexually have organs called gonads, the ovaries and testes, which produce eggs and sperm. These organs, which are different in males and females, originate from the same cells during the development of the embryo. As a general rule, the chromosomal sex of an embryo, which gets determined at fertilization, leads to the activation and repression of specific genes. This in turn, controls whether the cells that will form the gonads will differentiate to develop testes or ovaries.

Disruption of the key genes involved in the differentiation of the gonads can lead to fertility problems, and in some cases, it can cause the gonads to develop in the 'opposite' direction, resulting in a sex reversal. Identifying these genes is therefore essential to know how to maintain or restore fertility.

DMRT1 is a gene that drives the differentiation of gonadal cells into the testicular pathway in several species of animals with backbones, including species of fish, frogs and birds. However, its role in mammals – where testis differentiation is driven by a different gene called SRY – is not well understood. Indeed, when DMRT1 is disrupted in male humans it leads to disorders of sex development, while disrupting this gene in male mice causes infertility. To obtain more information about the roles of DMRT1 in mammalian species, Dujardin et al. disrupted the gene in a third species of mammal: the rabbit.

Dujardin et al. observed that chromosomally-male rabbits lacking DMRT1 developed ovaries instead of testes, showing that in rabbits, both SRY and DMRT1 are both required to produce testes. Additionally, this effect is similar to what is seen in humans, suggesting that rabbits may be a better model for human gonadal differentiation than mice are. Additionally, Dujardin et al. were also able to show that in female rabbits, lack of DMRT1 led to infertility, an effect that had not been previously described in other species.

The results of Dujardin et al. may lead to better models for gonadal development in humans, involving DMRT1 in the differentiation of testes. Interestingly, they also suggest the possibility that mutations in this gene may be responsible for some cases of infertility in women. Overall, these findings indicate that DMRT1 is a key fertility gene.

copies of the DMRT1 gene are required to induce testis determination. In ZW females and ZZ chickens harboring a non-functional copy, gonads differentiate as ovaries showing that sex determination is based on DMRT1 dosage (**Ioannidis et al., 2021**).

In mammals, where the sex-determining system is XX/XY, the TDF is the SRY gene (Sex-determining Region of the Y chromosome) carried by the Y chromosome. Based on the mouse species, DMRT1 does not appear to have retained a crucial function in testis determination since targeted deletion of Dmrt1 only affects post-natal testis function. In fact, DMRT1 has roles in both germ cells and supporting cells in the testis, and Dmrt1$^{-/-}$ males showed spermatogenesis failure with spermatogonia that did not undergo meiosis (**Matson et al., 2010**). However, specific knock-out of Dmrt1 in adult Sertoli cells led to their transdifferentiation into granulosa cells (**Matson et al., 2011**). Although DMRT1 is not required for testis determination in mice, it retained part of its function in adulthood when it is necessary to maintain Sertoli cell identity. In ovarian differentiation, FOXL2 (Forkhead family box L2) showed a similar function discrepancy between mice and goats as DMRT1 in the testis pathway. In the mouse, Foxl2 is expressed in female-supporting cells early in development but does not appear necessary for fetal ovary differentiation (**Uda et al., 2004**). On the contrary, it is required in adult granulosa cells to maintain female-supporting cell identity (**Ottolenghi et al., 2005**; **Uhlenhaut et al., 2009**). In other mammalian species, such as goats, FOXL2 was shown to be crucial for ovarian determination. Indeed, naturally observed in the PIS (Polled Intersex Syndrome) mutation (**Pailhoux et al., 2001**) or experimentally induced by genome editing in goats (**Boulanger et al., 2014**), FOXL2 loss-of-function led to female-to-male sex reversal with the early development of XX testes. Following FOXL2 absence of expression in the XX mutant gonads (XX PIS$^{-/-}$ or XX FOXL2$^{-/-}$), DMRT1 was up-regulated within days before increased SOX9 expression, which then directs the differentiation of Sertoli cells and the formation of testicular cords (**Elzaiat et al., 2014**). These observations in the goat suggested that DMRT1 could retain function in SOX9 activation and, thus, in testis determination in several mammals. In humans, a few

mutations affecting *DMRT1* have been described in patients presenting 46, XY DSD (Disorders of Sex Development) (*Chauhan et al., 2017*; *Ledig et al., 2012*; *Mello et al., 2010*). In particular, a heterozygous *de novo* point mutation in the *DMRT1* gene has been identified in a 46, XY individual with complete gonadal dysgenesis (*Murphy et al., 2015*), suggesting that DMRT1 and SRY may be involved in testicular determination.

To clarify DMRT1 functions in non-rodent mammals, we have chosen the rabbit model, where we generated a DMRT1 mutant line thanks to the CRISPR/Cas9 technology. Firstly, we characterized the DMRT1 expression in control gonads, showing that both XY and XX fetal gonads were expressing *DMRT1* before their sexual differentiation. In XY fetuses, *DMRT1* and *SRY* presented partially overlapping territory, and somatic cells expressing both of them harbored SOX9 expression and differentiated into Sertoli cells. Secondly, thanks to our CRISPR/Cas9 genetically modified rabbit model, we demonstrated that DMRT1 was required for testis differentiation since XY *DMRT1⁻/⁻* rabbits showed early male-to-female sex reversal with differentiating ovaries and complete female genitalia. However, germ cells failed to undergo meiosis, and follicles did not form in XY and XX *DMRT1⁻/⁻* mutant ovaries, leading to female infertility. Finally, we demonstrated that DMRT1 was a testis-determining factor in mammals and that it was also required for female fertility.

## Results

### *DMRT1* is expressed in genital crests of both sexes and just after *SRY* in XY developing testes

*DMRT1* expression pattern has already been reported by molecular analysis in the rabbit species from 14 days *post-coïtum* (d*pc*) to adulthood (*Daniel-Carlier et al., 2013*). We aimed to investigate further the location of the *DMRT1* expression during gonadal development, firstly at earlier stages of genital crest formation (12–13 d*pc*; *Figure 1A*) using *in situ* hybridization (ISH). *SRY* expression was already detected at 12 d*pc* and, as expected, was found only in the XY genital ridges, where it was restricted to the medullary part of the gonad (*Figure 1B*). In contrast, *DMRT1* was faintly expressed in the gonads of both sexes, in a few cells of the *medulla* under the coelomic epithelium (*Figure 1B*). At 12 d*pc*, only very few germ cells, expressing *POU5F1*, have completed their migration into the genital ridges (*Figure 1B*). Twenty-four hours later, at 13 d*pc*, the genital ridges had tripled in size in both sexes, and the territory of *SRY* expression increased within the XY developing testes (*Figure 1C*). The number of somatic cells expressing *DMRT1* was also strongly increased in both sexes, with few of them located in the coelomic epithelium (*Figure 1C*). In addition, more *POU5F1*-expressing germ cells were detected (5–12 per section instead of 1 or 2 at 12 d*pc*) (*Figure 1B, C*).

### SOX9 is detected in XY medullar cells co-expressing *SRY* and DMRT1

At 14 d*pc*, the SOX9 protein was immunodetected in a few cells located in the medullary part of the XY gonad (*Figure 2*). Numerous somatic cells of this region also expressed *SRY* and DMRT1 (*Figure 2*), and a few co-expressed SOX9 and DMRT1 simultaneously (*Figure 2—figure supplement 1*). In contrast, coelomic epithelial cells only expressed DMRT1 (*Figure 2*).

At 15 d*pc*, Sertoli cells that co-express *SRY*, DMRT1, and SOX9 began to be organized into embryonic cords (*Figure 2*). At this stage, coelomic epithelial cells expressed DMRT1 but were negative for *SRY* and SOX9. Furthermore, we observed an islet of cells expressing *SRY* and DMRT1 located in the mesonephros below the boundary with the gonad (*Figure 2*, dotted line). These cells expressed PAX8 (*Figure 2—figure supplement 2*) and could correspond to the recently described supporting-like cell population contributing to the *rete testis* in mice (*Mayère et al., 2022*). As in mice, these cells will express SOX9 at the latter stages (a few of them are already SOX9 positive at 15 d*pc*), but unlike mice, they express *SRY*.

From 16 to 18 d*pc*, the development of the testicular cords proceeded. At these two stages (16 and 18 d*pc*), *SRY*, DMRT1, and SOX9 were expressed only in the Sertoli cells, where *SRY* expression began to decrease from 18 d*pc* (*Figure 2*). No more DMRT1 expression could be seen in the coelomic epithelial cells, but the tunica albuginea begins to form (*Figure 2*), and consequently, the coelomic epithelium will become the surface epithelium.

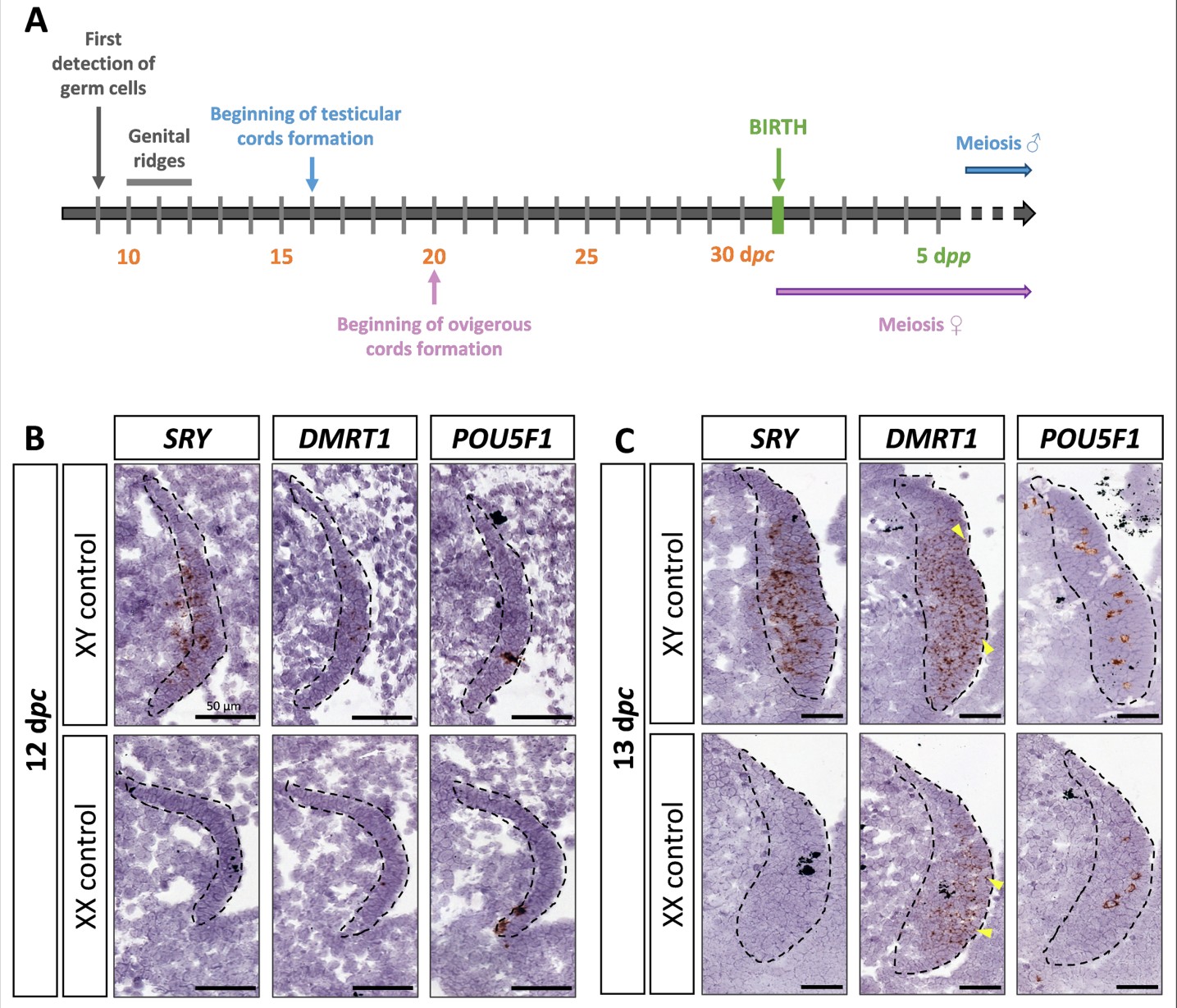

**Figure 1.** *SRY*, *DMRT1*, and *POU5F1* location during early gonadal development. (**A**) Key stages of gonadal development in rabbits with 31 days of gestation. Germ cells are first detected at 9 days *post-coïtum* (dpc), before the genital ridge formation, which occurs between 10 and 12 dpc. In XY gonads, testicular cords begin forming at 16 dpc, and germ cells enter meiosis a few months after birth. In XX gonads, the ovigerous cords appear at 20 dpc, and meiosis begins around birth. Location of *SRY*, *DMRT1*, and *POU5F1* by *in situ* hybridization (RNAscope technology) on XY and XX control gonads at (**B**) 12 dpc or (**C**) 13 dpc. Dotted line: developing genital crests. Yellow arrowheads: coelomic epithelial cells expressing *DMRT1*. Scale bar = 50 µm.

## Persistent expression of DMRT1 in XX gonadal somatic cells until ovigerous nest formation

As described above, *DMRT1* expression started at 12 dpc in the gonadal somatic compartment of both sexes (*Figure 1B*). In the female gonads, *DMRT1* remained expressed in all somatic cells, including those of the coelomic epithelium, until 16 dpc (*Figure 3A*). Interestingly, as in XY gonads, we observed PAX8-positive cells in XX gonads at 15 dpc (*Figure 2—figure supplement 1*). These cells could contribute to the formation of the *rete ovarii* as in mice (*Mayère et al., 2022*). At 18 dpc, *DMRT1* expression decreased but persisted in some cells located in the coelomic epithelium and just below it, where ovigerous nest formation occurred. Interestingly, the female *DMRT1*-antagonist

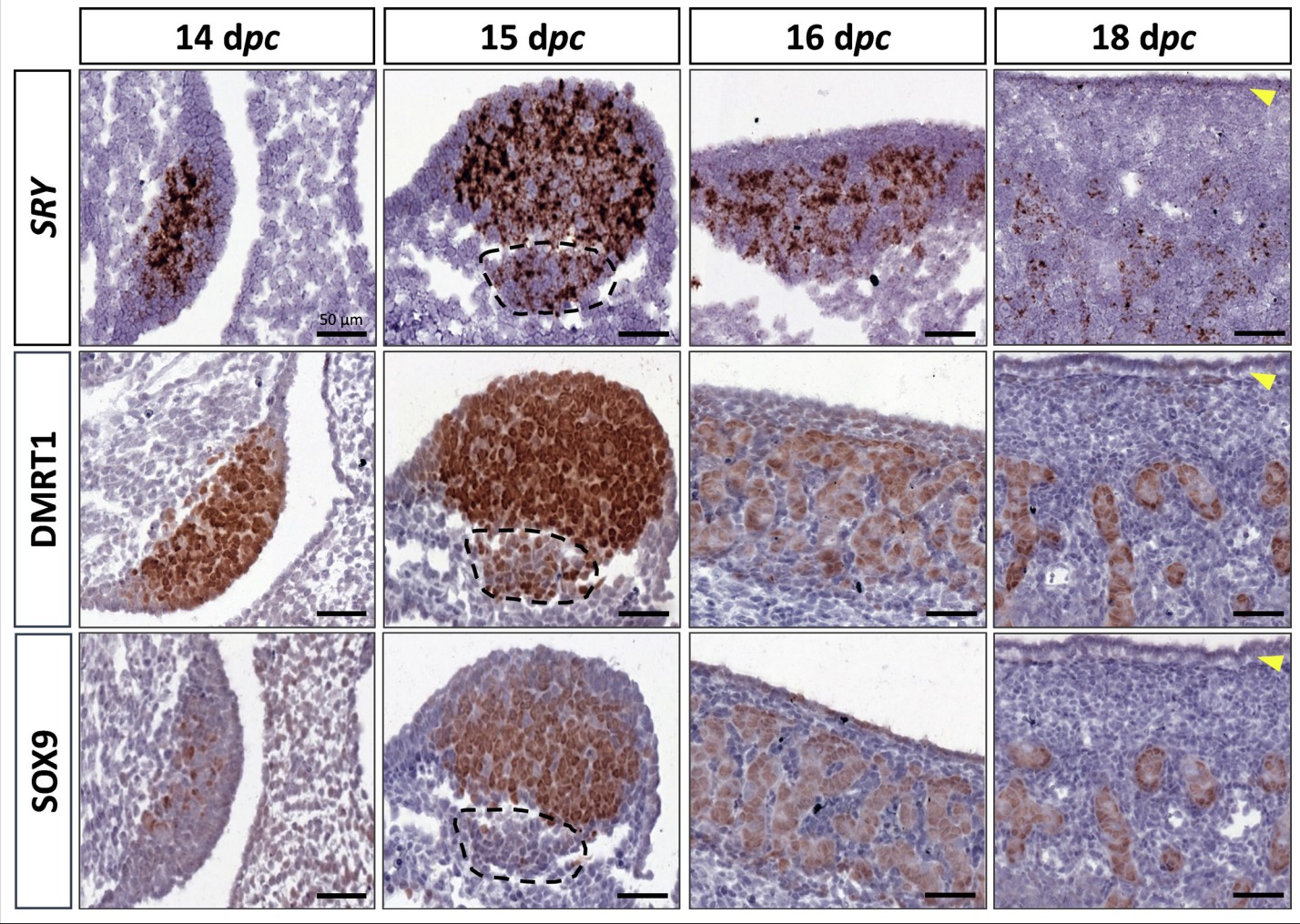

**Figure 2.** Somatic markers location during testis differentiation. Location of *SRY* by *in situ* hybridization (RNAscope technology), DMRT1, and SOX9 by immunohistochemistry on XY control testes from 14 to 18 d*pc*. The dotted line at 15 d*pc*: territory with cells expressing *SRY* and DMRT1 but not SOX9. Yellow arrowheads: tunica albuginea in formation. Scale bar = 50 µm.

The online version of this article includes the following figure supplement(s) for figure 2:

**Figure supplement 1.** DMRT1 and SOX9 co-location on 14 d*pc* XY control gonad.

**Figure supplement 2.** Identification of PAX8-positive cells in 15 d*pc* control gonads.

gene *FOXL2* began to be expressed between 16 and 18 d*pc* when *DMRT1* expression decreased (*Figure 3B*). Thereafter, at 20 d*pc*, *DMRT1* expression was limited in some somatic cells enclosed in nascent ovigerous nests where some germinal cells also began to be positive for DMRT1 (*Figure 3C* and *Figure 3—figure supplement 1*). At this stage, DMRT1-positive territory seems to overlap that of RSPO1 but not that of FOXL2 located in the loose conjunctive tissue around the ovigerous nests (*Figure 3C*).

## The testicular formation is impaired in *DMRT1* knock-out XY rabbits

To determine the role of DMRT1 in the rabbit species used as a non-rodent mammalian model, we engineered a *DMRT1* knock-out line using the CRISPR/Cas9 system with two RNA guides located in exon 3. The mutation carried by this line is a 47-bp duplication in sense, leading to a frameshift of the open reading frame and a premature stop codon (*Figure 4—figure supplement 1A*). This mutation does not affect *DMRT1* transcription but induces a total absence of protein as shown in post-natal gonads by western blot (*Figure 4—figure supplement 1B, C*). Thanks to this line, we first analyzed gonadal formation at 20 d*pc*, when the testis and ovary were distinguishable in control

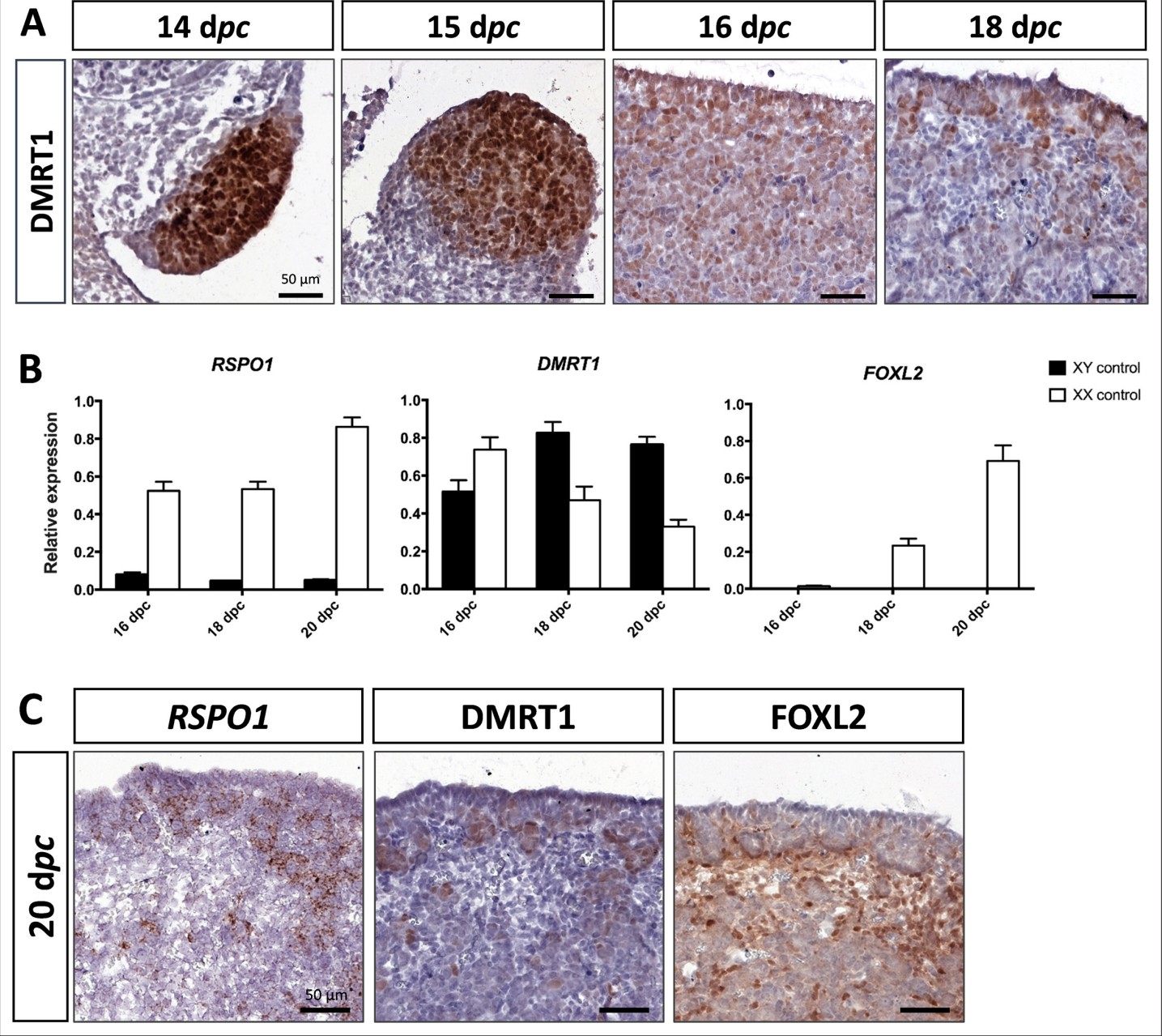

**Figure 3.** Somatic markers location and expression during ovarian differentiation. (**A**) Immunostaining of DMRT1 on XX control ovaries from 14 to 18 d*pc*. (**B**) Quantitative RT-PCR (RT-qPCR) analyses of *RSPO1*, *DMRT1*, and *FOXL2* expression from 16 to 20 d*pc* in control gonads of both sexes. The error bars correspond to the standard error of the mean (n=3-5) (**C**) *RSPO1 in situ* hybridization (RNAscope technology), immunostaining of DMRT1 and FOXL2 on 20 d*pc* control ovaries. Scale bar = 50 μm.

The online version of this article includes the following figure supplement(s) for figure 3:

**Figure supplement 1.** DMRT1 and POU5F1 co-detection in control gonads.

animals. Indeed, at this stage, testes appeared with well-formed seminiferous cords, and ovigerous nest formation was clearly in progress in the ovaries (***Figure 4A***). At 20 d*pc*, XY *DMRT1*⁻/⁻ gonads failed to engage testicular differentiation and appeared quite like control ovaries, but ovarian differentiation did not appear to be affected by the loss of *DMRT1* (***Figure 4A***). To better characterize the *DMRT1*⁻/⁻ gonads in XY and XX fetuses, we established the gonadal transcriptome by RNA-sequencing. Heatmap representation of the 3640 differentially expressed genes in at least one of the four genotypes (adjusted p-value <0.05 and |log2FC| > 1; ***Supplementary file 1***) was clustered into

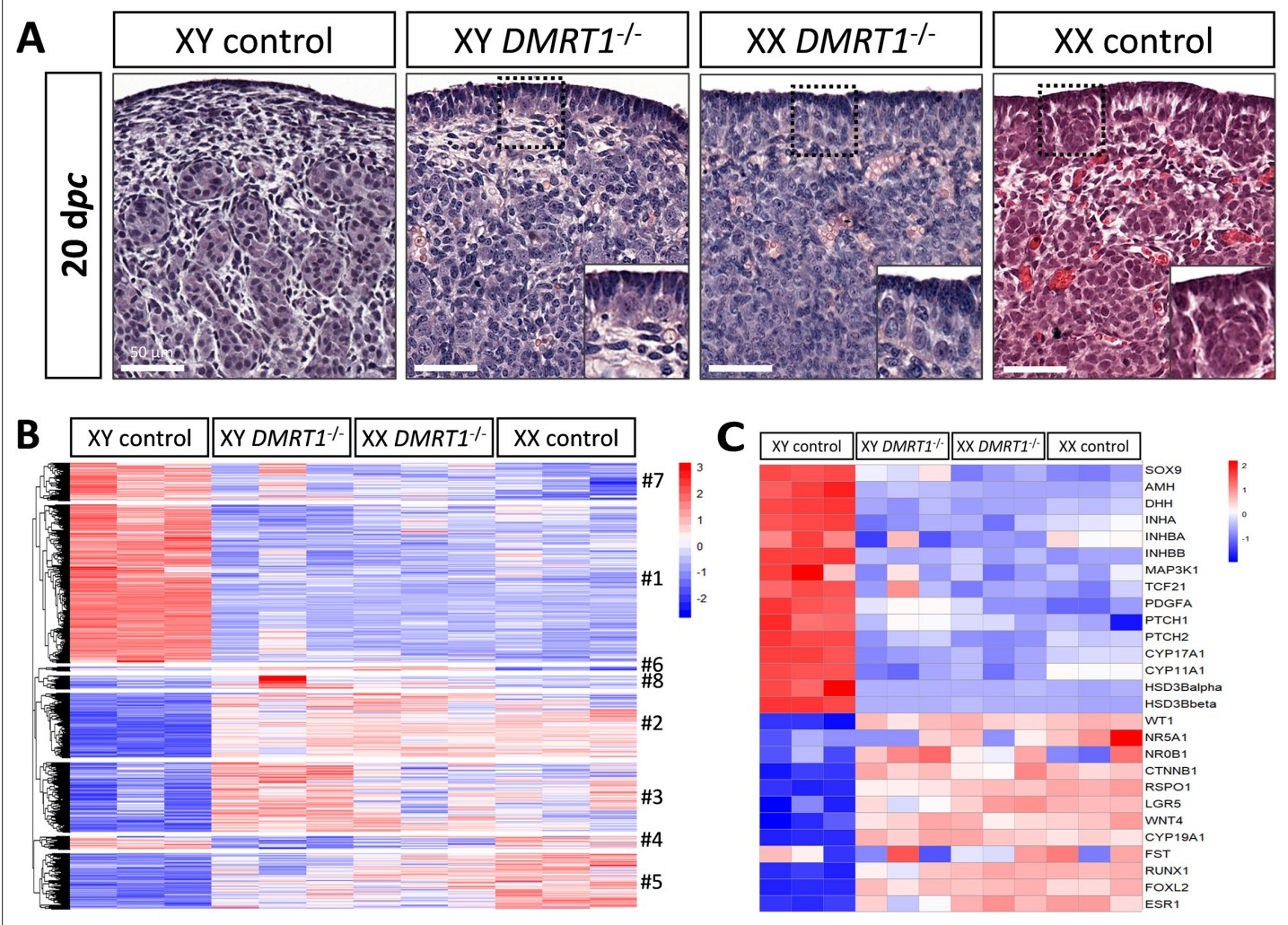

**Figure 4.** Ovarian-like morphology and transcriptomic signature of XY *DMRT1*−/− gonads at 20 d*pc*. (**A**) Hematoxylin and eosin staining of gonads sections from control and *DMRT1*−/− 20 d*pc* rabbits. The enlarged area shows the characteristic ovarian surface epithelium found on XY *DMRT1*−/− gonads. Scale bar = 50 µm. Heatmap representation of (**B**) 3460 deregulated genes (adjusted p-value <0.05 and |log2FC| > 1) or (**C**) 27 selected genes between XY control, XY *DMRT1*−/−, XX *DMRT1*−/−, and XX control at 20 d*pc*.

The online version of this article includes the following figure supplement(s) for figure 4:

**Figure supplement 1.** *DMRT1* mutation using CRISPR/Cas9 in rabbits.

eight groups (#1 to #8, *Figure 4B* and *Supplementary file 2*). Clusters #1 and #7 contained 1331 and 315 genes, respectively, which were preferentially expressed in XY control testes. Expression of these genes was decreased in XY *DMRT1*−/− gonads, harboring levels close to that of the female's ovaries (XX control or *DMRT1*−/−). On the other hand, clusters #2, #3, and #5 (537, 582, and 464 genes, respectively) were composed of genes preferentially expressed in XX control ovaries, and their expression was increased in XY *DMRT1*−/− gonads. Deep-sequencing transcriptomics confirmed the ovarian fate of XY *DMRT1*−/− gonads. The heatmap in *Figure 4C* also illustrates the expression for selecting some of the main genes involved in sex determination (*Figure 4C*).

Expression levels and patterns of the principal actors of gonadal differentiation were confirmed by quantitative RT-PCR (RT-qPCR), and the location of positive cells was achieved by immunohistochemistry.

As expected, *SOX9*, *AMH*, and *DHH* expression levels were decreased in XY *DMRT1⁻ᐟ⁻* gonads, remaining like those detected in control or *DMRT1⁻ᐟ⁻* XX ovaries, while *SRY* expression was enhanced in XY *DMRT1⁻ᐟ⁻* gonads (*Figure 5A*). Interestingly, we noticed a slight increase of SOX9-positive cells in XY *DMRT1⁻ᐟ⁻* gonads compared to XX control or mutant ovaries (*Figure 5C*). In contrast, *FOXL2* and *CYP19A1* expression were increased in XY *DMRT1⁻ᐟ⁻* gonads to similar levels to those detected in control or mutant ovaries (*Figure 5B*). By immunohistochemistry, we detected cells expressing FOXL2 in XY *DMRT1⁻ᐟ⁻* gonads (*Figure 5C*). Moreover, *RSPO1* expression was increased in XY *DMRT1⁻ᐟ⁻* gonads, but it remained lower than in control ovaries or in XX *DMRT1⁻ᐟ⁻* gonads. In the latter, the *RSPO1* expression was also lower than in control ovaries, suggesting a regulatory link between DMRT1 and RSPO1 in the female pathway (*Figure 5B*).

## Germ cells failed to engage meiosis in *DMRT1* mutant gonads

After the sex determination process and the first stages of gonad formation, *DMRT1⁻ᐟ⁻* gonads engage a female fate and differentiate as ovaries, whatever their sex-chromosome constitution, XX or XY. Whereas the *DMRT1* expression began at 18 d*pc* in the XY germinal lineage of control gonads and 20 d*pc* in XX (*Figure 3—figure supplement 1*), its expression was abolished in both somatic and germ cells in *DMRT1⁻ᐟ⁻* mutant gonads (*Figure 5C*). Although XX or XY *DMRT1⁻ᐟ⁻* gonads continue to develop as ovaries, most germ cells did not engage in the meiotic process. Indeed, in control ovaries at 3 days *post-partum* (d*pp*), most germ cells were in the zygotene stage, showing nuclei with highly condensed chromatin (*Daniel-Carlier et al., 2013*; *Figure 6*) and were positives for Ki67, showing their exit from the G0 phase of the cell cycle (*Figure 6—figure supplement 1*). In contrast, in *DMRT1⁻ᐟ⁻* gonads, few germ cells in the preleptotene stage were observed (*Figure 6*), and the majority did not express Ki67 but continued to express the pluripotency marker POU5F1 (*Figure 6—figure supplement 1*). Subsequently, the rupture of ovarian nests and the follicle formation did not occur in *DMRT1⁻ᐟ⁻* gonads. At 18 d*pp*, folliculogenesis had already started in control ovaries, where the first primordial follicles were visible in the deepest cortical part close to the *medulla* (*Figure 6*). In contrast, *DMRT1⁻ᐟ⁻* gonads seemed to be blocked at a pre-meiotic stage, and folliculogenesis failed to occur (*Figure 6*). In adults, *DMRT1⁻ᐟ⁻* gonads were reduced in size (*Figure 6—figure supplement 2*), no germ cells were detected, and some somatic cells evolved toward luteinized cells (*Figure 6*). Consequently, both XY and XX females were completely infertile in adulthood.

## Discussion

Our study gave new insights into the conservation of the sex-determination genetic cascade across evolution. Although the signal controlling this process could take different forms in metazoans, several downstream transcription factors involved in gonadal differentiation have been conserved throughout evolution. For instance, *SOX9*, well known in vertebrates as being essential for Sertoli cell differentiation (*Chaboissier et al., 2004*; *Foster et al., 1994*; *Qin and Bishop, 2005*; *Vidal et al., 2001*; *Wagner et al., 1994*), has a fruit fly ancestor, *Sox100B*, which was found to be necessary for testis development in *Drosophila* (*Nanda et al., 2009*). However, the most conserved sex-differentiating factor throughout evolution is DMRT1. Indeed, it has been maintained at the head of the sex determination cascade in reptiles (*Sun et al., 2017*), fishes (*Matsuda et al., 2002*), and birds (*Smith et al., 2009*). Nevertheless, its functions could have been reduced in mammals since testis differentiates in the absence of DMRT1 (*Dmrt1⁻ᐟ⁻*) in mice (*Raymond et al., 2000*). Our results highlight an evolutionary continuum of this gene in testis determination from birds to rabbits and non-rodent mammals in general. Interestingly, even *DMRT1* dosage sensibility has been conserved between chicken and rabbits since heterozygous XY *DMRT1⁺ᐟ⁻* male rabbits present secondary infertility with an arrest of spermatogenesis around 2 years of age (data not shown).

## DMRT1 position in the rabbit sex-determining cascade

As the early stages of gonadal differentiation in rabbits were not fully characterized, we first determined the expressional profiles of the major sex-determining genes. We observed that DMRT1 expression started at 12 d*pc*, at the early formation of genital crests, and it was first expressed in the somatic lineage of both sexes, as in mice (*Lei et al., 2007*; *Raymond et al., 1999*) or in humans (*Garcia-Alonso et al., 2022*). In the human fetal testis, *DMRT1* expression is co-detected with *SRY* in

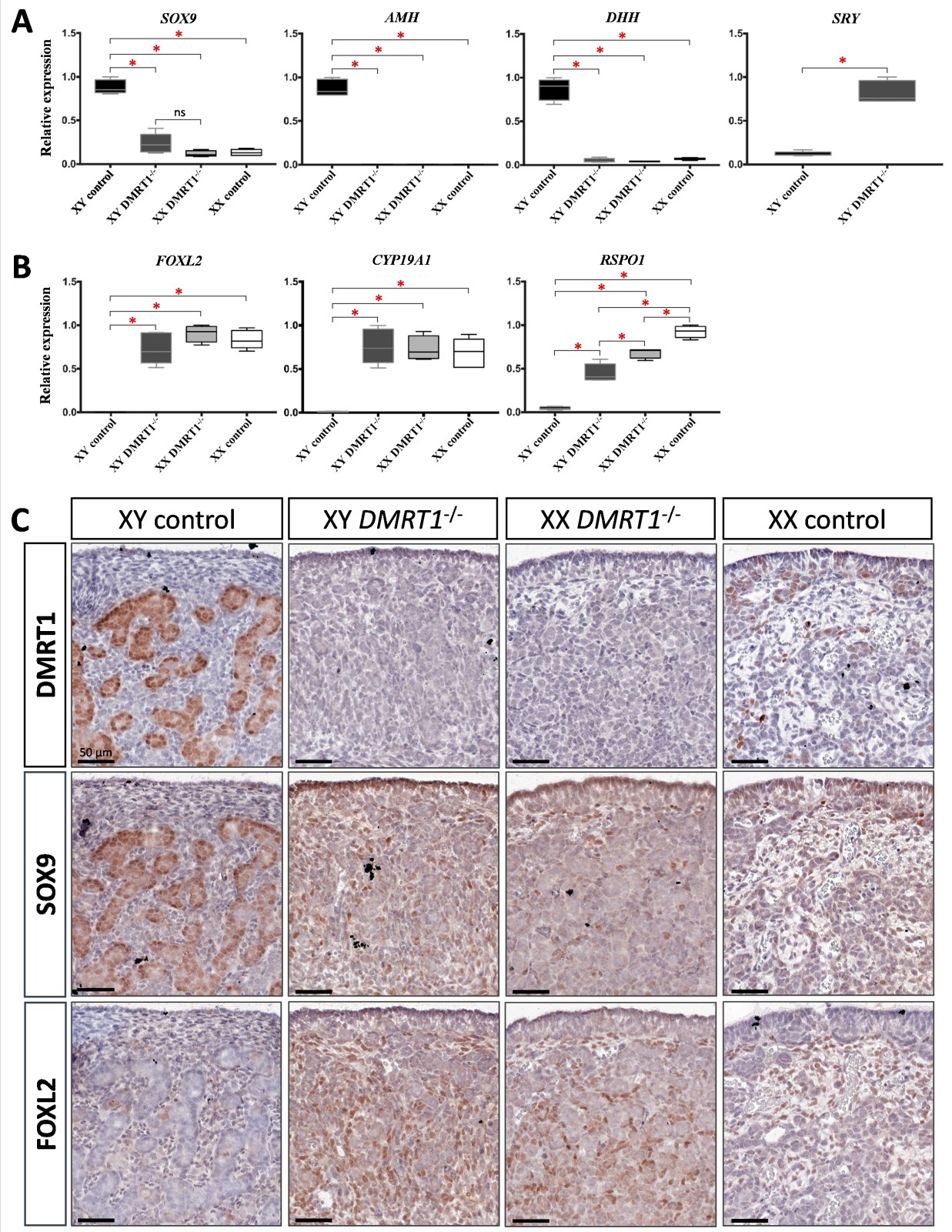

**Figure 5.** Somatic markers expression and location on control and *DMRT1⁻/⁻* gonads at 20 d*pc*. Quantitative RT-PCR (RT-qPCR) analyses of (**A**) testicular-related differentiation genes (*SOX9*, *AMH*, *DHH*, and *SRY*) or (**B**) ovarian-related differentiation genes (*FOXL2*, *CYP19A1*, and *RSPO1*) in XY control, XY *DMRT1⁻/⁻*, XX *DMRT1⁻/⁻*, and XX control gonads (*n* = 4–5) at 20 d*pc*. Statistical analyses were performed using the non-parametric Kruskal–Wallis test, followed by a pairwise permutation test: **p*-value <0.05; ns: non-significant. (**C**) Immunostaining of DMRT1, SOX9, and FOXL2 on XY control, XY *DMRT1⁻/⁻*, XX *DMRT1⁻/⁻*, and XX control gonad sections at 20 d*pc*. Scale bar = 50 µm.

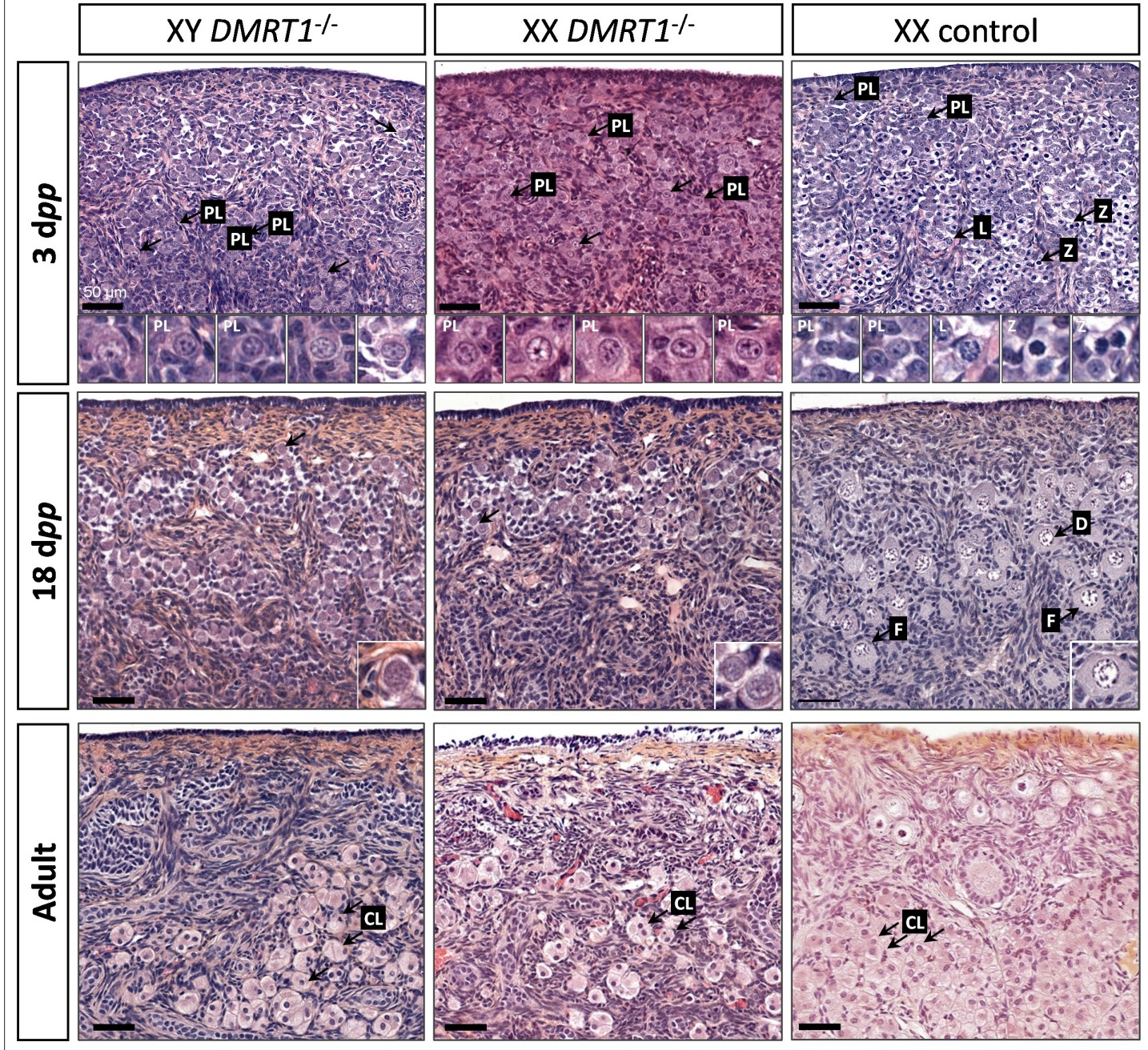

**Figure 6.** Evolution of gonadal morphogenesis in XY and XX *DMRT1⁻/⁻* rabbits. Hematoxylin and eosin staining of gonad sections from XY and XX *DMRT1⁻/⁻* gonads and XX control ovaries at 3 days *post-partum* (d*pp*), 18 d*pp*, and in adulthood (4–9 months). The enlargements for the first two panels correspond to the nuclei pointed by an arrow. PL: preleptotene stage; L: leptotene stage; Z: zygotene stage; D: diplotene stage; F: ovarian follicle; CL: luteal cells. Scale bar = 50 μm.

The online version of this article includes the following figure supplement(s) for figure 6:

**Figure supplement 1.** POU5F1 and Ki67 location on control and *DMRT1⁻/⁻* gonads at 3 d*pp*.

**Figure supplement 2.** Evolution of gonadal size in XY and XX *DMRT1⁻/⁻* rabbits.

early supporting gonadal cells, which become Sertoli cells following the activation of SOX9 expression (*Garcia-Alonso et al., 2022*). In mice, the *Dmrt1* expression starts at E10.5 in both somatic and germinal compartments. However, we showed that germline expression was shifted by 6–8 days compared to the somatic compartment in the rabbit male and female gonads, respectively. These differences are strongly related to the timing of gonadal development in rabbits – which is longer

than in mice – and therefore allows better visualization of the different processes. These sequential DMRT1 up-regulations according to cell type and sex also argue in favor of distinct *DMRT1* promoters as already described in rats (*Lei et al., 2009*). For the somatic XY compartment, *SOX9* expression appears at 14 d*pc* in cells expressing both *DMRT1* and *SRY*, suggesting that both factors are required for *SOX9* up-regulation. This led to the Sertoli cell differentiation and testicular cords formation from 15 d*pc*. In the developing ovary, we showed that *FOXL2* increases when DMRT1 expression starts to shift from somatic cells to germ cells. Moreover, our results suggested DMRT1 involvement in *RSPO1* up-regulation in the ovary.

## DMRT1 is required for testis determination in rabbits

In recent years, the advent of new genome editing technologies has made it possible to explore other animal models, such as the goat (*Boulanger et al., 2014*) or the rabbit (*Jolivet et al., 2022*), and enriching our knowledge on the conservation of ancestral genetic mechanisms in non-rodent mammals. In rabbits, the CRISPR-Cas9 technology allowed us to generate a null mutation of the *DMRT1* gene, leading to an absence of detectable protein at homozygosity. Thanks to this model, we could demonstrate that DMRT1 kept its leadership in sex determination also in mammals, where SRY stays the 'switch-on factor' for testis determination, as previously demonstrated in rabbits (*Song et al., 2017*). Very early in fetal life, XY fetuses expressing SRY but lacking DMRT1 (*DMRT1*$^{-/-}$) presented a male-to-female sex reversal. Although SRY expression was maintained in XY homozygous mutant gonads, the activation of *SOX9* expression was weak in the absence of DMRT1. Accordingly, a few cells expressing SOX9 protein were detectable, but SOX9 target genes expression were not activated in XY *DMRT1*$^{-/-}$ gonads. Thus, DMRT1 seems to be required for SRY action on its targets (i.e., *SOX9* gene activation) but also for SOX9 functions in the early fetal gonad. Interestingly, a recent study proposed that DMRT1 can act as a SOX9 pioneer factor in the post-natal testis for Sertoli cell identity maintenance (*Lindeman et al., 2021*). In rabbits, DMRT1 is required for SOX9 and SRY functions, and we hypothesize that DMRT1 might be a pioneer factor for both. In the differentiating genital crest, DMRT1 would be required to increase chromatin accessibility on specific sex-related regions, allowing SRY to bind and activate its targets and particularly the expression of *SOX9*. The crucial region for SRY binding was identified in mice more than 500 kb upstream of the *Sox9* transcription start site and named Enhancer 13 (*Gonen et al., 2018*). Conservation studies identified the homolog of Enhancer 13 in many mammalian species, including humans, cows, and rabbits, and DMRT1 consensus sites were predicted in all mammals examined except mice and rats (*Gonen et al., 2018*). In non-rodent mammals, DMRT1 might be required for chromatin remodeling on the Enhancer 13 region to enable SRY binding and *SOX9* expression since the beginning of testis differentiation. In the mouse, which evolved more rapidly, DMRT1 would no longer be necessary for SRY action because the chromatin state of the fetal supporting cells would be more permissive. This could also explain why DMRT1 does not exert any critical function in the fetal testis in mice (*Raymond et al., 2000*). In contrast, it is required for the action of SOX9 in the post-natal testis (*Lindeman et al., 2021*), where a sex-specific epigenetic signature was observed (*Garcia-Moreno et al., 2019*).

## DMRT1 is required for germ cell meiosis and female fertility

In addition to its functions in testis differentiation, DMRT1 also plays a crucial role in the female gonad. Indeed, germ cells did not undergo meiosis in *DMRT1*$^{-/-}$ ovaries, and in the absence of oocyte I, germ cell cysts do not break, compromising follicle formation and female fertility. This specific phenotype is highly similar to those observed in ZW chicken ovaries lacking DMRT1 (*Ioannidis et al., 2021*), but is quite different from those described in mice. Even though fewer follicles were observed in *Dmrt1*$^{-/-}$ mice ovaries, the female remains fertile (*Krentz et al., 2011*). Interestingly in humans, one case involving *DMRT1* in premature ovarian failure has been reported (*Bartels et al., 2013*).

In rabbit fetuses, DMRT1 expression was first detected in differentiating ovarian somatic cells, at least until *FOXL2* up-regulation. However, DMRT1 has also been observed in fetal germ cells from 20 d*pc* until meiosis proceeded after birth. Consequently, germ cell pre-meiotic arrest in *DMRT1*$^{-/-}$ XX gonads could result from *DMRT1* loss-of-function in the germinal or the somatic compartment or both. In the somatic compartment, the absence of DMRT1 in XX homozygous mutants did not seem to disturb the first steps of ovarian differentiation. Nevertheless, deep-sequencing transcriptomics revealed the dysregulated expression of a few genes involved in the WNT/beta-catenin pathway. In particular, *RSPO1*, a

positive regulator of the WNT signaling, was reduced, and DKK1, a negative regulator, was increased (*Supplementary file 1* and *Supplementary file 2*). These two events could have the effect of limiting the beta-catenin action in both somatic and germinal ovarian cells at the beginning of their differentiation. This pathway has proven to be crucial in mice to promote germ cell meiosis (*Le Rolle et al., 2021*). Nevertheless, it cannot be the main event explaining the pre-meiotic failure, and the functions of DMRT1 in germ cells are more certainly involved. In mice, DMRT1 was shown to be involved in *Stra8* up-regulation in female germ cells and was thus related to the meiotic process (*Krentz et al., 2011*). This regulatory action also seems to be done in close collaboration with the retinoic acid pathway (*Feng et al., 2021*). The sole action of DMRT1 on *STRA8* up-regulation cannot explain the phenotype observed in rabbits where the germ cell seems to be unable to leave their pluripotency stage. It has also been demonstrated in male mice that DMRT1 acts as a regulator of spermatogonia pluripotency by directly regulating different pluripotency-associated genes, including POU5F1 (*Krentz et al., 2009*; *Zhang et al., 2016*). This path is under exploration in our model in order to try to decipher further the critical role of DMRT1 in the germ line of both sexes.

# Materials and methods

## Key resources table

| Reagent type (species) or resource | Designation | Source or reference | Identifiers | Additional information |
|---|---|---|---|---|
| Biological sample (*Oryctolagus cuniniculus*) | Gonads | Hypharm | NZ1777 | New Zealand rabbits |
| Commercial kit | RNAscope kit | ACD | 322310 | 2.5HD assay-brown |
| Sequence-based reagent | RNAscope probe anti-DMRT1 | ACD | 410481 | XM_002708188.1 |
| Sequence-based reagent | RNAscope probe anti-SRY | ACD | 803191 | AY785433.1 |
| Sequence-based reagent | RNAscope probe anti-POU5F1 | ACD | 513271 | NM_001099957.1 |
| Sequence-based reagent | RNAscope probe anti-RSPO1 | ACD | 488231 | XM_002720657.2 |
| Antibody | Anti-DMRT1 (mouse monoclonal) | Santa Cruz | sc-377167 | IHC (1:500) IF (1:200) WB (1:100) |
| Antibody | Anti-SOX9 (rabbit polyclonal) | Francis Poulat | | IHC (1:500) IF (1:200) |
| Antibody | Anti-FOXL2 (rabbit polyclonal) | *Boulanger et al., 2014* | | IHC (1:500) |
| Antibody | Anti-POU5F1 (goat polyclonal) | Santa Cruz | sc-8628 | IHC (1:500) IF (1:200) |
| Antibody | Anti-PAX8 (rabbit polyclonal) | Proteintech | 10226-1-AP | IHC (1:6000) |
| Antibody | Anti-Ki67 (rabbit monoclonal) | Thermo Scientific | MA5-14520 | IHC (1:500) |
| Sequence-based reagent | SRY_F | This paper | PCR primers | TGCTTACACACCAGCCAAACA |
| Sequence-based reagent | SRY_R | This paper | PCR primers | TTCCTGGCCGCTCACTTTAC |
| Sequence-based reagent | DMRT1_F | This paper | PCR primers | GGAGCCTCCCAGCACCTTA |
| Sequence-based reagent | DMRT1_R | This paper | PCR primers | TGCATCCTGTACTGCGAACTCA |
| Sequence-based reagent | SOX9_F | This paper | PCR primers | GGCTCCGACACCGAGAATACAC |
| Sequence-based reagent | SOX9_R | This paper | PCR primers | GAACTTGTCCTCTTCGCTCTCCTT |
| Sequence-based reagent | CYP19A1_F | This paper | PCR primers | GGAAGAATGCATCGACTTGAGTT |
| Sequence-based reagent | CYP19A1_R | This paper | PCR primers | GGGCCCAAAACCAAATGGT |

*Continued on next page*

*Continued*

| Reagent type (species) or resource | Designation | Source or reference | Identifiers | Additional information |
|---|---|---|---|---|
| Sequence-based reagent | RSPO1_F | This paper | PCR primers | GCCCGCCTGGATACTTCGA |
| Sequence-based reagent | RSPO1_R | This paper | PCR primers | GGTGCAGAAGTTGTGGCTGAA |
| Sequence-based reagent | FOXL2_F | This paper | PCR primers | TTTCCCCTTTCCCCCATCTG |
| Sequence-based reagent | FOXL2_R | This paper | PCR primers | CTGAACCTTGCACCCAGCAT |
| Sequence-based reagent | AMH_F | This paper | PCR primers | GCTCATCCCCGAGACCTAC |
| Sequence-based reagent | AMH_R | This paper | PCR primers | CATCTTCAACAGCAGCACC |
| Sequence-based reagent | DHH_F | This paper | PCR primers | GCAATAAGTACGGGCTGCTG |
| Sequence-based reagent | DHH_R | This paper | PCR primers | GGCCAGGGAGTTATCAGCTT |
| Software | qBase+ | Biogazelle | | |
| Software | GraphPad Prism | GraphPad Software | | |

## Animals

New Zealand rabbits (NZ1777, Hypharm, Roussay, France) were bred at the SAAJ rabbit facility (Jouy-en-Josas, France). All experiments were performed with the approval of the French Ministry MENESR (accreditation number APAFIS#685 and #21451) and following the guidelines issued by the local committee for ethics in animal experimentation (COMETHEA, Jouy-en-Josas). All scientists working directly with the animals possessed an animal experimentation license delivered by the French veterinary services. Hormonal superovulation treatments and surgical embryo transfer procedures were performed as previously described (*Peyny et al., 2020*).

## Generation of mutant rabbits

Two guide RNAs were designed (http://crispor.trefor.net/) to target the third exon, as shown in *Figure 4—figure supplement 1A*. Embryos produced from superovulated females were injected at the single-cell stage with a mixture of the two sgRNAs (10 ng/µl each) and the Cas9mRNA (10 ng/µl) in the injection buffer. Injected embryos were implanted 3–4 hr after into the oviducts of anesthetized recipient rabbits via laparotomy. Details concerning the handling of females and embryos have been described elsewhere (*Peyny et al., 2020*).

Offspring were screened for the presence of InDel mutations using genomic DNA extracted from ear clips (*Jolivet et al., 2014*). Founders were detected by PCR using one set of primers (*Table 1*) surrounding the position of the targeted region in exon III (*Figure 4—figure supplement 1A*). The amplified fragment was sequenced (Eurofins Genomics, Courtaboeuf, France), and the mutation was deduced by comparing it with the sequence of a wild-type rabbit. The same set of primers was used for the routine screening of descendants. The presence/absence of the Y chromosome was deduced from the amplification of the SRY gene through PCR analyses (*Table 1*). In the present paper, mentions of the XY or XX genotype always refer to the PCR determination.

XY and XX $DMRT1^{+/-}$ rabbits were viable until adulthood and did not appear to have any diseases. $DMRT1^{-/-}$ mutants were obtained by crossing XY $DMRT1^{+/-}$ and XX $DMRT1^{+/-}$ animals.

## Histological and immunohistological analyses

Immediately after sampling, whole embryos or gonads were immersed in 4% paraformaldehyde (PFA) in phosphate-buffered saline (PBS) or Bouin's fixative. After 72–96 hr of fixation at 4°C, tissues were washed three times with PBS, and stored at 4°C in 70% ethanol until paraffin inclusions. Adjacent sections of 5 µm thick were processed using a microtome (Leica RM2245) and organized on Superfrost Plus Slides (J18000AMNZ, Epredia). Before staining or experiments, sections were deparaffinized and rehydrated in successive baths of xylene and ethanol at room temperature.

Hematoxylin–eosin–saffron (HES) staining was performed by the @Bridge platform (INRAE, Jouy-en-Josas, France) using an automatic Varistain Slide Stainer (Thermo Fisher Scientific).

ISH was performed using the RNAscope ISH methodology (ACD, Bio-Techne SAS, Rennes, France) when no reliable antibody could be used to characterize the target protein. Briefly, 5 µm sections

**Table 1.** Primers used for genotyping PCR or quantitative RT-PCR (RT-qPCR) analyses.

| Gene | Forward (5'–3') | Reverse (3'–5') |
| --- | --- | --- |
| Genotyping PCR | | |
| DMRT1 | TTTGAGCTGTGTCCCCAGAGT | ACCTCCCCAGAAGAAGAATCG |
| SRY | GTTCGGAGCACTGTACAGCG | GCGTTCATGGGTCGCTTGAC |
| RT-qPCR analyses | | |
| SRY | TGCTTACACACCAGCCAAACA | TTCCTGGCCGCTCACTTTAC |
| DMRT1 | GGAGCCTCCCAGCACCTTA | TGCATCCTGTACTGCGAACTCA |
| SOX9 | GGCTCCGACACCGAGAATACAC | GAACTTGTCCTCTTCGCTCTCCTT |
| CYP19A1 | GGAAGAATGCATCGACTTGAGTT | GGGCCCAAAACCAAATGGT |
| ESR1 | TCCTCATCCTCTCCCACATC | AGCATCTCCAGCAACAGGTC |
| RSPO1 | GCCCGCCTGGATACTTCGA | GGTGCAGAAGTTGTGGCTGAA |
| FOXL2 | TTTCCCCTTTCCCCCATCTG | CTGAACCTTGCACCCAGCAT |
| AMH | GCTCATCCCCGAGACCTAC | CATCTTCAACAGCAGCACC |
| DHH | GCAATAAGTACGGGCTGCTG | GGCCAGGGAGTTATCAGCTT |
| H2AFX | ACCTGACGGCCGAGATCCT | CGCCCAGCAGCTTGTTGAG |
| YWHAZ | GGGTCTGGCCCTTAACTTCTCT | AGCAATGGCTTCATCAAAAGC |
| SF1 (splicing factor 1) | GCTTCCGACTGCAAATTCCA | TCACCCAGTTCAGCCATGAG |

from PFA-fixed tissue were labeled using RNAscope 2.5HD assay-brown kit (322310, ACD) and 1000 nucleotides long probes designed and produced by the manufacturer (list of all synthesized probes used in *Table 2*). Brown labeling was observed as a visible signal, and hybridization was considered to be positive when at least one dot was observed in a cell.

Immunohistochemistry (IHC) was performed using the ABC amplification signal kit (PK-6100, Vector Laboratories) and DAB enzymatic reaction (SK-4100, Vector Laboratories). Briefly, the antigenic sites were unmasked with a citrate buffer (pH 6; H-3300, Vector Laboratories), and endogenous peroxidases were blocked with a 3% $H_2O_2$ solution (H1009, Sigma-Aldrich). Sections were then permeabilized with 1× PBS, 1% bovine serum albumin (A7906, Sigma-Aldrich), and 0.2% saponin (7395, Merck) and incubated overnight at 4°C with primary antibodies (*Table 3*). Following PBS washes, sections were incubated with biotinylated secondary antibodies (*Table 3*). After ABC kit incubation and DAB revelation, hematoxylin staining was briefly performed to visualize the whole tissue.

Immunofluorescence (IF) was performed using Tyramide SuperBoost kit for primary rabbit antibody (B40944, Thermo Fisher) as recommended by the manufacturer. Other secondary antibodies used are listed in *Table 3*.

All stained sections were scanned using a 3DHISTECH panoramic scanner at the @Bridge platform (INRAE, Jouy-en-Josas, France).

## Total RNA extraction and RT-qPCR

Immediately after sampling, 16–20 d*pc* gonads were snap-frozen in liquid nitrogen and stored at −80°C until extraction. Total RNAs were isolated using Trizol reagent (15596018, Life Technologies),

**Table 2.** Synthesized probes used for *in situ* hybridization.

| Gene name | RNAscope probe catalog number | Transcript accession number |
| --- | --- | --- |
| DMRT1 | 410481 | XM_002708188.1 |
| SRY | 803191 | AY785433.1 |
| POU5F1 | 513271 | NM_001099957.1 |
| RSPO1 | 488231 | XM_002720657.2 |

**Table 3.** List of antibodies used for immunohistochemistry (IHC), immunofluorescence (IF), or western blot (WB).

| | Antibody name | Reference | Dilution |
|---|---|---|---|
| **Primary antibodies** | Mouse monoclonal to DMRT1 | sc-377167 (Santa Cruz) | 1/500 (IHC); 1/200 (IF); 1/100 (WB) |
| | Rabbit polyclonal to SOX9 | Francis Poulat | 1/500 (IHC); 1/200 (IF) |
| | Rabbit polyclonal to FOXL2 | *Boulanger et al., 2014* | 1/500 (IHC) |
| | Goat polyclonal to POU5F1 | sc-8628 (Santa Cruz) | 1/500 (IHC); 1/200 (IF) |
| | Rabbit polyclonal to PAX8 | 10336-1-AP (Proteintech) | 1/6000 (IHC) |
| | Rabbit monoclonal to Ki67 | MA5-14520 (Thermo Scientific) | 1/500 (IHC) |
| **Biotinylated secondary antibodies (IHC)** | Horse anti-rabbit | BA-1100 (Vector Laboratories) | 1/200 |
| | Anti-mouse | Included in M.O.M. kit (BMK-2202, Vector Laboratories) | 1/200 |
| | Horse anti-goat | BA-9500 (Vector Laboratories) | 1/200 |
| | Poly HRP-conjugated goat anti-rabbit | B40944 (Invitrogen) | No diluted |
| | DyLight 488 goat anti-mouse | 072-03-18-06 (KPL) | 1/200 |
| **Secondary antibodies (IF)** | Alexa Fluor 594 chicken anti-goat | A21468 (Life technologies) | 1/200 |

purified with the RNeasy Micro kit (74004, QIAGEN) following the manufacturer's instructions, and then DNAse treated (1023460, QIAGEN). RNAs were quantified with a Qubit Fluorometric Quantification kit (Q32852, Life Technologies).

Reverse transcription of 50–100 ng RNAs using the Maxima First-Strand cDNA Synthesis Kit (K1641, Thermo Scientific) was down. qPCR with diluted cDNA was performed in duplicate for all tested genes with the Step One system (Applied Biosystems) and Fast SYBR Green Master Mix (4385612, Applied Biosystems). *H2AFX* and *YWHAZ* or *SF1* (Splicing Factor 1) were used as the reference genes to normalize the results with qBase$^+$ software (Biogazelle NV, Ghent, Belgium). The sequences of the primers used are listed in *Table 1*.

For each experiment, values were plotted using GraphPad Prism Software (GraphPad Software Inc, La Jolla, CA, USA). Statistical analyses of data from 20 d*pc* control and *DMRT1*$^{-/-}$ gonads were performed under R studio software. Because of the small number of samples in each group, comparisons were made using the Kruskal–Wallis rank sum test followed by pairwise permutation *t*-tests (1000 permutations, p-value adjusted with the Benjamini–Hochberg method).

## Nuclear proteins extraction and western blot

Gonads from newborns (1–3 days *post-partum*) rabbits were collected and snap-frozen in liquid nitrogen and then stored at −80°C. Frozen gonads were crushed in liquid nitrogen using a mortar. Powdered tissue samples were immediately resuspended in homogenization buffer (10 mM HEPES pH 7.7; 25 mM KCl; 2 mM Sucrose; 0.5 mM EGTA pH 8; 0.15 mM Spermin; 0.5 mM Spermidin; 0.5 mM Dithiothreitol (DTT); 2 mM Benzamidin; 0.5 mM Phenylmethylsulfonyl fluoride (PMSF); cOmplete, Mini, EDTA-free Protease Inhibitor Cocktail (Roche, 118361700001)) plus 0.3% IGEPAL (3021, Sigma-Aldrich). After centrifugation for 15 min at 4°C and 3500 rpm, supernatants containing cytosolic proteins were stored at −80°C. Pellets were centrifugated for 1 hr at 4°C and 12,700 rpm and then resuspended with [C-NaCl] buffer (20 mM HEPES pH 7.7; 1.5 mM MgCl$_2$; 0.2 mM EDTA; 25% glycerol; 0.5 mM PMSF; 0.5 mM DTT; 2 mM Benzamidin; cOmplete, Mini, EDTA-free Protease Inhibitor Cocktail). NaCl was added, lysates were rotated for 1 hr at 4°C and then centrifugated for 30 min at 4°C and 12,700 rpm. Supernatants containing nuclear extracts were collected, and the amount of protein was determined by the Bradford method.

Protein (20 µg of each sample) was separated on 4–15% polyacrylamide gel (456-1083, Bio-Rad) and then transferred into a polyvinylidene difluoride membrane. The membrane was blocked in 4% milk (Difco Skim Milk #232100 diluted in PBS-Tween 2%) and incubated overnight with primary antibodies mouse anti-DMRT1 (*Table 3*) or mouse anti-beta Actin (1/5000; GTX26276, Genetex). After washes, the membrane was incubated for 1 hr at room temperature with the secondary antibody

anti-mouse IgG peroxidase-conjugated (1/500; A5906, Sigma-Aldrich). The revelation was performed using Pierce ECL Plus Western Blotting Substrate (32312, Thermo Fisher), and the signal was observed with the Chemi-Doc Touch Imaging System (Bio-Rad). For rehybridization, the membrane was stripped for 10 min in Restore Western Blot stripping buffer (21059, Thermo Fisher).

### RNA-sequencing and bioinformatics analysis

Total RNAs were extracted from control and $DMRT1^{-/-}$ rabbit gonads at 20 d$pc$ ($n$ = 3 for each phenotype and each sex). Total RNA quality was verified on an Agilent 2100 Bioanalyser (Matriks, Norway), and samples with a RIN >9 were made available for RNA-sequencing. This work benefited from the facilities and expertise of the I2BC High-throughput Sequencing Platform (https://www.i2bc.paris-saclay.fr/sequencing/ng-sequencing/ Université Paris-Saclay, Gif-sur-Yvette, France) for oriented library preparation (Illumina Truseq RNA Sample Preparation Kit) and sequencing (paired-end 50–35 bp; NextSeq500). More than 37 million 50–35 bp paired-end reads per sample were generated. Demultiplexing was done (bcl2fastq2-2.18.12), and adapters were removed (Cutadapt1.15) at the I2BC High-throughput Sequencing Platform. Only reads longer than 10 pb were used for analysis. Quality control of raw RNA-Seq data was processed by FastQC v0.11.5.

Reads were mapped on all the genes of a better-annotated rabbit genome. Indeed, we improved the current reference rabbit transcriptome (OryCun2.0; *Oryctolagus cuniculus*, Ensembl version 106). For this purpose, we extended the 5′ and 3′-UTRs of genes using rabbit gonad RNA-seq data available in public databases (https://www.ncbi.nlm.nih.gov/bioproject/PRJEB26840). In addition, the annotation and for some of them, sequences of 22 marker genes of gonadal differentiation missing or wrong in OryCun2.0 was added or fixed to this genome assembly. Then, after mapping with STAR version 2.5.1b (*Dobin et al., 2013*), reads were counted using FeatureCounts version 1.4.5 (*Liao et al., 2014*). Data normalization and single-gene level analyses of differential expression were performed using DESeq2 (*Love et al., 2014*). Differences were considered to be significant for Benjamini–Hochberg adjusted p-values <0.05, and absolute fold log2FC >1 (*Benjamini and Hochberg, 1995*). RNA-seq data were deposited via the SRA Submission portal (https://www.ncbi.nlm.nih.gov/sra/PRJNA899447), BioProject ID PRJNA899447.

### Acknowledgements

The authors would like to thank Patrice Congar, Gwendoline Morin, and all the staff of the facility (SAAJ, INRAE, Jouy-en-Josas, France) for the care of the rabbits, Erwana Harscoët and Nathalie Daniel for injecting the embryos, Laurent Boulanger for the sgRNAs design, Nathalie Daniel-Carlier for the mutation characterization, Julie Rivière and Marthe Vilotte (UMR GABI, INRAE, Jouy-en-Josas, France) for their assistance on the histological platform (@Bridge platform) and the access to the virtual slide scanner, Namya Mellouk for her contribution to the dissection of the gonads, Simon Herman (Université Paris-Saclay, France) for his contribution to the improvement of the rabbit genome annotation during his master's internship in our team (DGP, UMR BREED, INRAE, Jouy-en-Josas, France), and Andrew Crawford (Academic Writing Center, Centralesupélec, France) for the English proofreading. Francis Poulat kindly provided the SOX9 antibody. We acknowledge the sequencing and bioinformatics expertise of the I2BC High-throughput sequencing facility, supported by France Génomique (funded by the French National Program 'Investissement d'Avenir' ANR-10-INBS-09). We are grateful to the genotoul bioinformatics platform Toulouse Occitanie (Bioinfo Genotoul, France, https://doi.org/10.15454/1.5572369328961167E12) for providing computing and storage resources. This work was supported by ANR (Agence Nationale de la Recherche) grants (RNA-SEX: ANR-19-CE14-0012; ARDIGERM: ANR-20-CE14-0022). ED was supported by the ANR RNA-SEX and the PHASE department of INRAE.

## Additional information

### Funding

| Funder | Grant reference number | Author |
|--------|------------------------|--------|
| Agence Nationale de la Recherche | ANR-19-CE14-0012 | Eric Pailhoux Emilie Dujardin |
| Agence Nationale de la Recherche | ARDIGERM: ANR-20-CE14-0022 | Eric Pailhoux |
| INRAE, France's National Research Institute for Agriculture, Food and Environment | | Emilie Dujardin |

The funders had no role in study design, data collection, and interpretation, or the decision to submit the work for publication.

### Author contributions

Emilie Dujardin, Investigation, Methodology, Writing – original draft, Writing – review and editing; Marjolaine André, Aurélie Dewaele, Francis Poulat, Methodology, Writing – review and editing; Béatrice Mandon-Pépin, Anne Frambourg, Dominique Thépot, Luc Jouneau, Investigation, Methodology, Writing – review and editing; Geneviève Jolivet, Conceptualization, Methodology, Writing – review and editing; Eric Pailhoux, Maëlle Pannetier, Conceptualization, Supervision, Funding acquisition, Writing – original draft, Writing – review and editing

### Author ORCIDs

Emilie Dujardin  https://orcid.org/0000-0002-8189-8742
Béatrice Mandon-Pépin  http://orcid.org/0000-0002-5424-0822
Francis Poulat  http://orcid.org/0000-0003-2070-1296
Maëlle Pannetier  https://orcid.org/0000-0002-5826-619X

### Ethics

All experiments were performed with the approval of the French Ministry MENESR (accreditation number APAFIS#685 and #21451) and following the guidelines issued by the local committee for ethics in animal experimentation (COMETHEA, Jouy-en-Josas). All scientists working directly with the animals possessed an animal experimentation license delivered by the French veterinary services.

Reviewer #1 (Public Review): https://doi.org/10.7554/eLife.89284.3.sa1
Reviewer #2 (Public Review): https://doi.org/10.7554/eLife.89284.3.sa2
Reviewer #3 (Public Review): https://doi.org/10.7554/eLife.89284.3.sa3
Author Response https://doi.org/10.7554/eLife.89284.3.sa4

## Additional files

### Supplementary files

• Supplementary file 1. List of DEGs in *DMRT1*$^{-/-}$ gonads (XY and XX) compared to control gonads (XY and XX) (adjusted p-value <0.05 and |log2FC| > 1). List of deregulated genes (DEGs) between KO-XY vs Control-XY gonads (sheet 1), KO-XY vs KO-XY gonads (sheet 2), KO-XX vs Control-XX gonads (sheet 3), and Control-XY vs Control-XX gonads (sheet 4). The gene name of DEGs was based on their annotation or human homology (*Craig et al., 2012*).

• Supplementary file 2. Clustering and expression values (TPM, transcripts per million) of the 3460 DEGs. Cluster membership of the 3460 DEGs with their expression data (TPM) according to the four genotypes (XY control, XY DMRT1$^{-/-}$, XX DMRT1$^{-/-}$, XX control).

• MDAR checklist

### Data availability

RNA-seq data were deposited via the SRA Submission portal, BioProject ID PRJNA899447.

The following dataset was generated:

| Author(s) | Year | Dataset title | Dataset URL | Database and Identifier |
|---|---|---|---|---|
| Dujardin E, Andre M, Dewaele A, Mandon-Pépin B, Poulat F, Frambourg A, Thépot D, Jouneau L, Jolivet G, Pailhoux E, Pannetier M | 2023 | RNA-seq of Oryctolagus cuniculus | https://www.ncbi.nlm.nih.gov/sra/PRJNA899447 | NCBI Sequence Read Archive, PRJNA899447 |

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
