## [Editor Report · eLife assessment]

In this **important** study, the rabbit was used as a non-rodent mammalian model to show that DMRT1 has a testicular promoting function as it does in humans. The experiments are meticulous and **compelling**, and the arguments are clear and **convincing**. These results may explain the gonadal dysgenesis associated with mutations in human DMRT1 and highlight the need for mammalian models other than mice to better understand the process of gonadal sex determination in humans.

---

## [Referee Report · Reviewer #1 (Public Review)]

DMRT1 is essential in testis development in different species. While Dmrt1 is the testis-determining factor in chicken and deletion encompassing this gene lead to gonadal dysgenesis in human, the role of DMRT1 in testis development remains to be clarified. Despite an early expression of Dmrt1 in the mouse gonad and a potential function as a pioneer factor, DMRT1 is only required for the maintenance of the Sertoli cell identity in the postnatal testis. The use of a new animal model could provide new insights into the role of this factor in humans. Here the authors have generated a knockout model of DMRT1 in rabbits. They show that the XY mutant gonads differentiate as ovary indicating that DMRT1 is required for testis differentiation in rabbits. In addition, most of the germ cells remain pluripotent as evidenced by the maintenance of POU5F1 in both XY and XX mutant gonads. These are very important results potentially explaining gonadal dysgenesis associated with the DMRT1 locus in disorders of sex development in humans.

The experiments are meticulous and convincing. I find the arguments of the authors about the role of DMRT1 in germ cells in addition to its function in Sertoli cell differentiation, both comprehensible and compelling. Clearly, this is an important insight in sex determination and gametogenesis.

---

## [Referee Report · Reviewer #2 (Public Review)]

It is well known that DMRT proteins and more specifically, DMRT1 plays a key role in the sex determination processes of many species. While DMRT1 has been shown to be critical for the sex determination of fish, birds, and reptiles, it seems less crucial at the sex determination stages of the mice. It is important though for adult sex maintenance in mice.

Unlike its minor role in mouse sex determination, it seems that variants in DMRT1 in humans cause 46, XY DSD and sex reversal.

The paper by Dujardin et al. is a beautiful study that provides an answer to this long-lasting discrepancy of the difference between the two common mammal species: human and mouse. It is a really nice example of how working with other mammal species, like the rabbit, could serve as a nice model for understanding mammalian sex determination.

In this study the researchers first described the expression patterns of DMRT1 in the rabbit XY and XX gonads throughout the window of sex determination.

They then used CRISPR/Cas9 to generate DMRT1 KO rabbits and analysed the phenotype in XY and XX rabbits. They show that XY rabbits present with complete XY male-to-female sex reversal, very similar to what observed in human 46, XY DSD patients (but not the mice model). They further show that in the XY sex reversed gonads, germ cells fail to enter meiosis. They next analysed XX gonads and while there is no major effect on sex determination (as expected), the germ cells in these ovaries fail to enter meiosis, highlighting the critical role that DMRT1 has in germ cells.

I think it is really important that we start to embrace other mammal models that are not the mouse as we find many instances that the mouse is not the optimal system for understanding human sex determination.

The study is well explained and presented. The data is clear, and the paper is fluent to read.

---

## [Referee Report · Reviewer #3 (Public Review)]

This manuscript deals with the sex-related gene, DMRT1, showing that is has a testis-promoting function in the rabbit. Loss-of-function studies the mouse and human, DMRT1 has a role in testis maintenance after birth, although forced expression in mouse can induce testis formation.

The authors used CRISPR/Cas9 genome editing to generate DMRT1-/- rabbit embryos. The gonads of these embryos developed as ovaries. Interestingly, in addition Y-linked SRY, DMRT1 is required for timely up-regulation of SOX9 during Sertoli cell differentiation in the male gonad. This is quite different to the situation in mouse, where Dmrt1 is not required in the testis until after birth (and Sry induced up-regulation of Sox9 hence does not require Dmrt1).

The work adds to the field of sex determination by further broadening our understanding of the DMRT1 gene and the evolution of gonadal sex determination.

In the Discussion section, it is suggested that DMRT1 could act as a pioneering factor to allow SRY action upon Sox9 in the rabbit model. The data show that DMRT1 may be more central to testis formation in mammals than previously considered. The work supports the notion that our understanding that the genetics of gonadal development (and indeed development more generally) should not rest solely on findings in the mouse.

---

## [Author Response]

The following is the authors’ response to the original reviews.

**Reviewer #1 (Recommendations For The Authors):**
Some sentences need to be clarified and some additional data and references could be added.1. Line 18SRY is the sex-determining geneSRY is the testis-determining gene is more accurate as described in line 44

Modification done

1. Line 50Despite losing its function in early testis determination in mice, DMRT1 retained part of this function in adulthood when it is necessary to maintain Sertoli cell identity.Losing its function is misleading. The authors describe firstly that Dmrt1 has no obvious function in embryonic testis development but is critical for the maintenance of Sertoli cells in adult mice. The wording "losing its function in early testis" is confusing. Do the authors mean that despite the expression of Dmrt1 in early testis development, the function of Dmrt1 seems to be restricted to adults in mice? A comparison between the testis and ovary should be more cautious since GarciaAlonso et al (2022) have shown that the transcriptomics of supporting cells between humans and mice is partly different.

That’s what we thought, and the sentence has been changed as follow: “Although DMRT1 is not required for testis determination in mice, it retained part of its function in adulthood when it is necessary to maintain Sertoli cell identity.” (line 51 to 53)

1. Line 78XY DMRT1-/- rabbits showed early male-to-female sex reversal.Sex reversal indicates that there is no transient Sertoli cell differentiation that transdifferentiate into granulosa cells. This brings us to an interesting point. In the case of reprogramming, the transient Sertoli cells can produce AMH leading to the regression of the Mullerian ducts. In humans, some 9pdeleted XY patients have Mullerian duct remnants and feminized external genitalia. This finding indicates early defects in testis development.Is there also feminized external genitalia in XY Dmrt1−/− rabbits. Can the authors comment on the phenotype of the ducts?

We proposed to add “and complete female genitalia” at the end of the following sentence:“Secondly, thanks to our CRISPR/Cas9 genetically modified rabbit model, we demonstrated that DMRT1 was required for testis differentiation since XY DMRT1-/- rabbits showed early male-tofemale sex reversal with differentiating ovaries and complete female genitalia.” (line 77 to 80)

Indeed, since the first stage (16 dpc) where we can predict the sex of the individual by observing its gonads during dissection, we always predict a female sex for XY DMRT1 KO fetuses. It is only genotyping that reveals an XY genotype. At birth, our rabbits are sexed by technicians from the facility and again, but now based on the external genitalia, they always phenotype these rabbits as female ones. In these XY KO rabbits, the supporting cells never differentiate into Sertoli, and ovarian differentiation occurs as early as in XX animals. Thus, these animals are fully feminized with female internal and external genitalia. Most of 9p-deleted patients are not homozygous for the loss-offunction of DMRT1, and the remaining wild-type allele could explain the discrepancy between KO rabbits and humans.

1. Line 53In the ovary, an equivalent to DMRT1 was observed since FOXL2 (Forkhead family box L2) is expressed in female supporting cells very early in development.Can the authors clarify what is the equivalent of DMRT1, is it FOXL2? DMRT1 heterozygous mutations result in XY gonad dysgenesis suggesting haploinsufficiency of DMRT1. However, to my knowledge, there is no evidence of haploinsufficiency in XX babies. Thus can we compare testis and ovarian genetics?

We agree, the term “equivalent” is ambiguous, and we changed the sentence as follows: “In ovarian differentiation, FOXL2 (Forkhead family box L2) showed a similar function discrepancy between mice and goats as DMRT1 in the testis pathway. In the mouse, Foxl2 is expressed in female supporting cells early in development but does not appear necessary for fetal ovary differentiation. On the contrary, it is required in adult granulosa cells to maintain female-supporting cell identity.” (line 53 to 56)

Regarding reviewer 2's question on haploinsufficiency in humans: the patient described in Murphy et al., 2015 is an XY individual with complete gonadal dysgenesis. But, it has been shown that the mutation carried by this patient leads to a dominant-negative protein, equivalent to a homozygous state (Murphy et al., 2022).

For FOXL2 mutation in XX females, haploinsufficiency does not affect early ovarian differentiation (no sex reversal) but induces premature ovarian failure.

We agree with the reviewer, we cannot compare testis and ovarian genetics considering two different genes.

1. Line 55In mice, Foxl2 does not appear necessary for fetal ovary differentiation (Uda et al., 2004), while it is required in adult granulosa cells to maintain female-supporting cell identity (Ottolenghi et al., 2005). The reference Uhlenhaut et al (2009) reporting the phenotype of the deletion of Foxl2 in adults should be added.

The reference has been added.

1. Line 64Lindeman et al (2021) have shown that DMRT1 can act as a pioneer factor to open chromatin upstream and Dmrt1 is expressed before Sry in mice (Raymond et al, 1999, Lei, Hornbaker et al, 2007). Whereas additional factors may compensate for the absence of Dmrt1, these results suggest that DMRT1 is also involved in Sox9 activation.

Dmrt1 is indeed expressed before Sry/Sox9 in the mouse gonad. However, no binding site for DMRT1 could be observed at Sox9 enhancer 13 in mice. This does not support a role for DMRT1 in the activation of Sox9 expression in this species. Furthermore, in Lindeman et al 2021, the authors clearly state that DMRT1 acts as a pioneering factor for SOX9 only after birth. It does not appear to have this role before. One of the explanations put forward is that the state of chromatin is different during fetal development in mice: chromatin is more permissive and does not require a factor to facilitate its opening. This hypothesis is based in particular on the description of a similar chromatin profile in the precursors of XX and XY fetal supporting cells, where many common regions display an open structure (Garcia-Moreno et al., 2019). Once sex determination and differentiation are established, a sex-specific epigenome is set up in gonadal cells. Chromatin remodeling agents are then needed to regulate gene expression. We hypothesize that in non-murine mammals such as rabbits, the state of gonadal cell chromatin would be different in the fetal period, more repressed, requiring the intervention of specific factors for its opening, such as DMRT1.

1. Figure 1Most of the readers might not be familiar with the developmental stages of the gonad in rabbits. A diagram of the key stages in gonad development would facilitate the understanding of the results.

Thank you, it has been added in Figure 1.

1. Figure 2Arrowheads are difficult to spot, could the authors use another color?

Done

1. Line 117: can the authors comment on the formation of the tunica albuginea? Do the epithelial cells acquire some specific characteristics?

The formation of the tunica albuginea begins with the formation of loose connective tissue beneath the surface epithelium of the male gonad. The appearance of this tissue is concomitant with the loss of expression of DMRT1 in the cell of the coelomic epithelium. Our interpretation is that the contribution of the cells from the coelomic epithelium and their proliferation stops when the tunica begins to form because the structure of the tissue beneath the epithelium change, and the cellular interactions between the epithelium and the tissue below remain disrupted. By contrast, these interactions persist in the ovary until around birth for ovigerous nest formation.

1. The first part of the results described DMRT1 expression in rabbits. With the new single-cell transcriptomic atlas of human gonads, it would be important to describe the pattern of expression in this species. This could be described in the introduction in order to know the DMRT1 expression pattern in the human gonad before that of the rabbit.

A comment on the expression pattern of DMRT1 in human fetal gonads has been added in the discussion section: “In the human fetal testis, DMRT1 expression is co-detected with SRY in early supporting gonadal cells (ESCGs), which become Sertoli cells following the activation of SOX9 expression (Garcia-Alonso et al., 2022) » (line 222 to 224)

1. Figure 3 supplement 3Dotted line: delimitation of the ovarian surface epithelium. Could the authors check that there is adotted line?

Done

1. Figure 5 and Line 186Quantification is missing such as the % of germ cells, % of meiotic germ cells.

Quantification is not easy to realize in rabbits because of the size and the elongated shape of the gonad. Indeed, it’s difficult to be sure that both sections (one from WT, the other from KO) are strictly in a similar region of the gonad and that the section is perfectly longitudinal or not. See also our answer to reviewer 3 (point 7) on this aspect. Actually, we are trying to make a better characterization of this XX phenotype and to find a marker of the pre-leptotene/leptotene stage susceptible to work in rabbits (SYCP3 will be the best, but we encountered huge difficulties with different antibodies and even RNAscope probe!). So actually, the most convincing indirect evidence of this pre-meiotic blockage (in addition to HE staining at 18 dpp in the new Figure 6) is the persistence of POU5F1 (pluripotency), specifically in the germinal lineage of KO XX and XY gonads. In addition to the new figure supplement 5, we can show you in Author response image 1: (i) the gonadal section at a lower magnification, where it is evident that there is a big difference between WT and KO germ cell POU5F1-stainings; and (ii) POU5F1 expression from a bulk RNA-seq realized the day after birth at 1 dpp where the difference is also transcriptionally very clear.

**Author response image 1. sa4fig1:** 

1. Line 186,E is missing at preleptoten

Added

1. Figure supplement 7.A magnification of the histology of the gonads is missing.

This figure is only for showing the gonadal size, and there are the same gonads as in the new Figure 6. So, the magnification is represented in Figure 6.

15)DiscussionLine 201SOX9, well known in vertebrates,

The references of the human DSD associated with SOX9 mutations are missing. Thank you, references have been added.

1. Line 286One of the targets of WNT signaling is Bmp2 in the somatic cells and in turn, Zglp1, which is required for meiosis entry in the ovary as shown by Miyauchi et al (2017) and Nagaoka et al (2020). Does the level of BMP pathway vary in DMRT1 mutants?

At 20 dpc, the expression level of BMP2 in XY and XX DMRT1 mutants gonads is similar to the one of XX control which is lower than in XY control (see the TMP values from our RNA-seq in Author response image 2).

**Author response image 2. sa4fig2:** 

**Reviewer #2 (Recommendations For The Authors):**
Here are my minor comments:1. Line 106- You mention that coelomic epithelial cells only express DMRT1. Please add an arrow to highlight where you refer to.

Done

1. Line 112: In mice, the SLCs also express Sox9 but not Sry apart from Pax8. You mention here that the SLCs are expressing SRY and DMRT1 in addition to PAX8. Could you perhaps explain the difference? Please refer to that in the results or discussion.

We add a new sentence at the end of this paragraph on SLCs: “As in mice, these cells will express SOX9 at the latter stages (few of them are already SOX9 positive at 15 dpc), but unlike mice, they express SRY.” (line 114 to 115)

We already have collaborations with different labs on these SLC cells, and we will certainly come back later on this aspect, remaining slightly off-topic here.

1. Could you please explain why did you chose to target Exon 3 of DMRT1 and not exons 1-2 which contain the DM domain? Was it to prevent damaging other DMRT proteins? Is there an important domain or function in Exon 2?

Our choice was mainly based on technical issues (rabbit genome annotation & sgRNA design), but also we want to avoid targeting the DM domain due to its strong conservation with other DMRT genes. Due to the poor quality of the rabbit genome, exons 1 and 2 are not well annotated in this species. We have amplified and sequenced the region encompassing exons 1 & 2 from our rabbit line, but the software used for sgRNA design does not predict good guides on this region. The two best sgRNAs were predicted on exon 3, and we used both to obtain more mutated alleles.

1. Your scheme in Supp Figure 4 is not so clear. It is not clear that the black box between the two guides is part of Exon 3 (labelled in blue).

The scheme has been improved.

1. Did you only have 1 good founder rabbit in your experiment? Why did you choose to work with a line that had duplication rather than deletion?

Very good point! In the first version of this paper, we’d try to explain the long (around 2 years) story of breeding to obtain the founder animal. Here it is:

During the genome editing process, we generate 6 mosaic founder animals (5 males and 1 female), then we cross them with wild-type animals to isolate each mutated allele in F1 offspring used afterward to establish and amplify knockout lines. Unexpectedly, we observe a very slow ratio of mutated allele transmission (5 on 129 F1 animals), and only one mutated allele has been conserved from the unique surviving adult F1 animal. It consists of an insertion of the deleted 47 bp DNA fragment, flanked by the cutting sites of the two RNA guides used with Cas9.

The main hypothesis to explain this mutation event is that in the same embryonic cell, the deletion occurs on one allele then the deleted fragment remains inserted into the other allele. Under this scheme, the embryonic cell carries a homozygous DMRT1 knockout genotype, albeit heterogeneous, with a deleted allele (del47) and the present allele (insertion of a 47 bp fragment leading to an in sense duplication). This may explain the very low frequency of transmission since all germ cells carrying a homozygous DMRT1-/- genotype will probably not be able to enter the meiotic process as suggested by our results on XX and XY DMRT1-/- ovaries. Finally, and under this hypothesis, the way we obtained this unique founder animal remains a mystery!

1. Figure 4- real-time data- where does it say what is a,b,c,d of the significance? It should appear on the figure itself and not elsewhere.

Modification done.

1. If I understand correctly, you were able to get the rabbits born and kept to adulthood (you show in supp figure 7 their gonads). What was the external phenotype of these rabbits? Did the XY mutant gonads have the internal and external genitals of a female (oviduct, uterus, vagina etc.)?

See our answer to Reviewer 1 on this question (point 3).

1. Line 20: It is more correct to write 46, XY DSD rather than XY DSD

Modification done.

1. Line 21: you can remove the "the" after abolished

Modification done.

1. Line 31: consider replacing the first "and" by "as well as" since the sentence sounds strange with two "and".

Modification done.

1. Line 212- Please check with the eLife guidelines if they allow "data not shown" in the paper.

This is unspecified.

**Reviewer #3 (Recommendations For The Authors):**
The following points should be addressed.1. The in situ's in Fig 1 and 2 are very clear. Fig 1 and Fig 2, In situ hybridisation in tissue sections, it looked like DMRT1 could be expressed in some cells where SRY mRNA is absent @ E13.5dpc and 14.5 dpc. Do you think this is real, or maybe Sry is turned off now in those cells?

Based on the results of in situ hybridizations, DMRT1 appears to be expressed by both coelomic epithelium and genital crest medullar cells in a pattern that is actually broader than that of SRY. Moreover, in rabbits, SRY expression seems to start in the medulla of the genital ridge rather than in the surface epithelium, as described in mice (see Figure 1 at 12 and 13 dpc). Nevertheless, more detailed analyses are needed to ensure the lineage of cells expressing SRY and/or DMRT1, such as single-cell RNAseq at these key stages of sexual determination in rabbits (from 12 to 16 dpc).

1. It is curious that SRY expression is elevated in the DMRT1 KO (Knockout) rabbit gonads. Does this suggest feedback inhibition by DMRt1, or maybe indirect via effect on Sox9 (as I believe Sox9 feeds back to down-regulate Sry in mouse, for example).

The maintenance of SRY expression in the DMRT1 -/- rabbit testis seems to be linked to the absence of SOX9 expression. We believe that, as in mice, SOX9 would down-regulate SRY (even if, in rabbits, SRY expression is never completely turned off).

1. I suggest the targeting strategy and proof of DMRT1 knockout by sequencing etc. be brought out of the suppl. Data and shown as a figure in the text.

See also our answer to reviewer 2 (point 5). It has needed huge efforts to obtain these DMRT1 mutated rabbit line, and of course, it constitutes the basis of the study. But regarding the title and the main message of the article, we are not convinced that the targeting strategy should be moved into the main text.

1. Unless there are limitations imposed by the journal, I also feel that Suppl Fig 5 (the immunostaining) deserves to be in the paper text too. The Fig showing loss of DMRt1 by immunostaining is important.

We include the figure supplement 5 in the main text. So, Figure 4E and figure supplement 5 have been combined into a new Figure 5.

1. The RT-qPCR data should have the statistics clarified on the graphs. e.g., it is stated that, although Sox9 mRNA is clearly down, there is a slight increase compared to control on KO XX gonads. Is this statistically significant? Figure legend states that the Kruskal-Wallis test is used, and significance is shown by letters. This is unclear. It would be better to use the more usual asterisks and lines to show comparisons.

Modification done.

1. Reference is made to DMRT1+/- rabbits having aberrant germ cell development, pointing to a dosage effect. This is interesting. Does the somatic part of the gonad look completely normal in the het knockouts?

DMRT1 heterozygous male rabbits have a phenotype of secondary infertility with aging, and we are trying now to better characterize this phenotype. The problem is complex because, as we cannot carry out conditional KO, it remains difficult to decipher the consequence of DMRT1 haploinsufficiency in the Sertoli cells versus the germinal ones. Anyway, the somatic part is sufficiently normal to support spermatogenesis since heterozygous males are fertile at puberty and for some months thereafter.

1. Can the authors indicate why meiotic markers were not used to explore the germ cell phenotype? It would be advantageous to use a meiotic germ cell marker to definitely show that the germ cells do not enter meiosis after DMRT1 loss. (Not just H/E staining or maintenance of POU). Example SYCP3, or STRA8 (as pre-meiotic marker) by in situ or immunostaining. Even though no germ cells were detected in adult KO gonads.

The expression of pre-meiotic or meiotic markers is currently under study in DMRT1 -/- females. Transcriptomic data (RNA-seq) are also being analyzed. We are preparing a specific article on the role of DMRT1 in ovarian differentiation in rabbits. We felt it was important to reveal the phenotype observed in females in this first article, but we still need time to refine our description and understanding of the role of DMRT1 in the female.

1. What future studies could be conducted? In the Discussion section, it is suggested that DMRT1 could act as a pioneering factor to allow SRY action upon Sox9. How could this be further explored?

To explore the function of DMRT1 as a pioneering factor, it now seems necessary to characterize the epigenetic landscapes of rabbit fetal gonads expressing or not DMRT1 (comparison of control and DMRT1-/- gonads). Two complementary approaches could be privileged: the study of chromatin opening (ATAC-seq) and the analysis of the activation state of regulatory regions (CUT&Tag). The study of several histone marks, such as H3K4me3 (active promoters), H3K4me1 (primed enhancers), H3K27ac (enhancers and active promoters), and H3K27me3 (enhancers and repressed promoters), would be of great interest. However, these techniques are only relevant for gonads that can be separated from the adjacent mesonephros, which is only possible from the 16 dpc stage in rabbits. To perform a relevant analysis at earlier stages, a "single-nucleus" approach such as ATAC-seq singlenucleus or multi-omic single-nucleus combining ATAC-seq and RNA-seq could be used.